# Plasma and Peritoneal Fluid Fibronectin and Collagen IV Levels as Potential Biomarkers of Endometriosis

**DOI:** 10.3390/ijms232415669

**Published:** 2022-12-10

**Authors:** Damian Warzecha, Julia Załęcka, Grzegorz Mańka, Mariusz Kiecka, Michał Lipa, Robert Spaczyński, Piotr Piekarski, Beata Banaszewska, Artur Jakimiuk, Tadeusz Issat, Wojciech Rokita, Jakub Młodawski, Maria Szubert, Piotr Sieroszewski, Grzegorz Raba, Kamil Szczupak, Tomasz Kluz, Marek Kluza, Mirosław Wielgoś, Łukasz Ołdak, Anna Leśniewska, Ewa Gorodkiewicz, Piotr Laudański

**Affiliations:** 11st Department of Obstetrics and Gynaecology, Medical University of Warsaw, 02-091 Warsaw, Poland; 2Angelius Provita Hospital, 40-611 Katowice, Poland; 3Division of Infertility and Reproductive Endocrinology, Department of Gynaecology, Obstetrics and Gynaecological Oncology, Poznan University of Medical Sciences, 61-701 Poznan, Poland; 4Department of Obstetrics and Gynaecology, Institute of Mother and Child in Warsaw, 01-211 Warsaw, Poland; 5Department of Obstetrics and Gynaecology, Central Clinical Hospital of the Ministry of Interior, 02-507 Warsaw, Poland; 6Collegium Medicum, Jan Kochanowski University in Kielce, 25-369 Kielce, Poland; 7Clinic of Obstetrics and Gynaecology, Provincial Combined Hospital in Kielce, 25-736 Kielce, Poland; 8Department of Gynaecology and Obstetrics, Medical University of Lodz, 90-419 Lodz, Poland; 9Department of Surgical Gynaecology and Oncology, Medical University of Lodz, 90-419 Lodz, Poland; 10Department of Fetal Medicine and Gynaecology, Medical University of Lodz, 90-419 Lodz, Poland; 11Clinic of Obstetrics and Gynaecology in Przemysl, 37-700 Przemysl, Poland; 12Department of Gynaecology and Obstetrics, University of Rzeszow, 35-310 Rzeszow, Poland; 13Department of Gynaecology, Gynaecology Oncology and Obstetrics, Institute of Medical Sciences, Medical College of Rzeszow University, 35-310 Rzeszow, Poland; 14Bioanalysis Laboratory, Faculty of Chemistry, University of Bialystok, 15-245 Bialystok, Poland; 15OVIklinika Infertility Center, 01-377 Warsaw, Poland; 16Women’s Health Research Institute, Calisia University, 62-800 Kalisz, Poland; 17Department of Obstetrics, Gynaecology, Gynecology and Gynaecological Oncology, Medical University of Warsaw, 02-091 Warsaw, Poland

**Keywords:** endometriosis, fibronectin, collagen IV, plasma, peritoneal fluid

## Abstract

Laparoscopy as a diagnostic tool for patients with suspected endometriosis is associated with several potentially life-threatening complications. Therefore, it is imperative to identify reliable, non-invasive biomarkers of the disease. The aim of this study was to analyse the concentrations of fibronectin and type IV collagen in peritoneal fluid and plasma to assess their role as potential biomarkers in the diagnosis of endometriosis. Fibronectin and collagen IV protein levels were assessed by surface plasmon resonance imaging (SPRi) biosensors with the usage of monoclonal antibodies. All patients enrolled in the study were referred for laparoscopy for the diagnosis of infertility or chronic pelvic pain (n = 84). The study group included patients with endometriosis confirmed during surgery (n = 49). The concentration of fibronectin in the plasma (329.3 ± 98.5 mg/L) and peritoneal fluid (26.8 ± 11.1 μg/L) in women with endometriosis was significantly higher than in the control group (251.2 ± 84.0 mg/L, 7.0 ± 5.9 μg/L). Fibronectin levels were independent of endometriosis stage (*p* = 0.874, *p* = 0.469). No significant differences were observed in collagen IV levels (*p* = 0.385, *p* = 0.465). The presence of elevated levels of fibronectin may indicate abnormalities in cell–ECM signalling during the course of endometriosis, and may be a potential biomarker for early detection.

## 1. Introduction

Endometriosis is a common gynaecological disease defined as the presence of endometrial tissue outside the uterine cavity, i.e., within the peritoneum and visceral organs. It affects up to 15% of women of reproductive age [1,2]. The symptoms of the disease are often mild or moderate; however, they can be associated with dysmenorrhoea, dyspareunia, chronic pain, and even infertility [3,4,5]. All these factors significantly reduce quality of life [6]. Non-specific symptoms of endometriosis often result in a disregard of the problem by patients and clinicians, which in turn is associated with a diagnostic delay of 7–9 years [7,8]. Therefore, many researchers are looking for reliable and non-invasive biomarkers for the disease [9,10,11].

Although many studies have been carried out on the aetiology of endometriosis, it has not been fully elucidated, and it remains unclear why the disease affects only a few women. It is well known that the mere presence of endometrial cells does not guarantee the onset of the disease [12]. According to the research, genetic, environmental, and microenvironmental factors—including a number of epigenetic and hormonal disorders—significantly predispose the development of the disease [13,14,15]. The pathogenesis of endometriosis is significantly affected by a disturbed immune balance associated with many inflammatory markers, such as immune cells, chemokines, cytokines, metalloproteinases, and miRNAs [16,17,18,19]. They are responsible for the development of ectopic lesions, their invasion, and accompanying angiogenesis [20,21,22,23]. Consequently, the disease is characterised by chronic inflammation, leading to fibrosis and adhesion formation.

It appears that the extracellular matrix (ECM) may contribute to the development of endometriosis due to its modulating properties for cell migration, proliferation, differentiation, and development. It is a three-dimensional network of molecules that provides cells with a complex microenvironment [24]. It is often tissue-specific, but each contains common protein classes, such as glycoproteins, collagens, elastins, and proteoglycans. The last three have structural functions [25]. Collagens are the most abundant part of the ECM, with most forming supramolecular assemblies. They ensure mechanical strength and are directly responsible for signal transmission to the cell. The cross-link density of the fibres is responsible for their tensile strength [26,27]. In developing tissues, intricate mechanisms create networks of collagen fibrils with specific structures and properties. Impairment of these processes ultimately leads to mechanical damage to the tissues and sometimes to fibrosis [28]. To date, more than 20 types of collagens have been characterised, each with different functions and structures. Type IV collagen is the main component of the basement membrane, which ensures the integrity and exchange of substances between epithelial cells and the environment [29]. Due to its functions, it is possible that collagen IV participates in the invasion of ectopic endometrial cells into the wall of the peritoneal cavity, marking their migration path. Glycoproteins serve as linkers that stabilise the ECM. The dominant glycoprotein occurring in organised structures is fibronectin [30]. Fibronectin fibres form a network responsible for connecting adjacent cells and binding soluble ECM molecules, such as growth factors [31]. Cell–ECM binding is essential for the integrity of information pathways and structural support. They are among the earliest proteins formed during tissue development and wound healing. They form the basis for further assembly of the matrix on which collagen and many other particles are deposited [32]. Abnormalities in the concentration of these proteins may translate into the impairment of signalling pathways between cells and create pathological conditions for the development of endometriosis.

Surface plasmon resonance imaging (SPRi) is an emerging, direct, and ‘label-free’ technique that has growing potential in the development of diagnostic biosensors. It involves a plasmon resonance-imaging-supporting metal surface coupling light energy with an electromagnetic field on cells and surface-associated fibronectin. It is characterised by a very sensitive measurement of the refractive index, which makes it ideal for the quantification of molecules, such as proteins [33]. Applying the stationary SPRi version in a model investigation and in the determination of different biomarkers in real clinical samples has demonstrated that this technique is suitable for use without preliminary analyte preconcentration or signal enhancement [34]. An additional advantage of this innovative technique is that it does not require any markers [35]. A growing number of clinical applications of SPRi have been shown in a recent review [36].

To date, type IV collagen has not been tested for its usefulness in detecting endometriosis; however, a few studies have indicated increased adhesion of endometrial cells to this protein [37]. However, research on fibronectin has yielded inconclusive results; therefore, further trials on larger groups of patients are necessary [38]. The aim of this study was to investigate the levels of fibronectin and collagen IV using SPRi biosensors in the plasma and peritoneal fluid to strengthen their position as potential non-invasive biomarkers of endometriosis.

## 2. Results

The study group included 46 patients with endometriosis confirmed during laparoscopy, and 35 controls. The baseline characteristics of the groups are presented in Table 1.

In the study group, the mean intensity of chronic pelvic pain within 12 months preceding laparoscopy was 7 (NRS scale). The mean age at the onset of the first symptoms of pelvic pain was 24.8 years (SD = 5.2), while the mean age at the time of final diagnosis of endometriosis was 29.7 (SD = 5.4). According to the American Society of Reproductive Medicine (ASRM) classification, 15 patients (30.6%) were classified as stage I, 7 as stage II (14.3%), 18 as stage III (36.7%), and 9 as stage IV (18.4%).

Table 2 presents plasma and peritoneal fluid concentrations of fibronectin and collagen IV among patients with and without endometriosis. In both groups, the concentrations of fibronectin and collagen IV were normally distributed. A statistically significant difference in fibronectin concentrations was demonstrated—both in the peritoneal fluid and in the plasma. There were no significant differences for collagen IV in both specimens.

In patients with endometriosis, the mean concentration of fibronectin in the serum was 329.3 [mg/L] whereas in the peritoneal fluid it was 26.8 [μg/L]. We found a significant correlation between plasma and peritoneal fibronectin levels in the endometriosis group (*p* = 0.001).

Among the samples from the study group, we did not observe any significant differences in fibronectin and collagen IV concentrations, depending on the severity of endometriosis. The data are presented in Table 3.

In multiple logistic regression for fibronectin and type IV collagen in the peritoneal fluid and plasma for the prediction of endometriosis, any of the assessed factors (age, phase of the cycle, or coexisting fertility impairments) achieved statistical significance (*p* < 0.05). None of the previously mentioned factors correlated with the peritoneal or plasma levels of fibronectin and type IV collagen. We did not observe any significant correlation between the plasma fibronectin concentration and the pelvic pain scale. The results of logistic regression for endometriosis are presented in Table 4 and Figure 1 separately for each predictor (OR with 95% CI and a *p*-value).

Figure 2, Figure 3 and Figure 4 present the receiver operating characteristic (ROC) curve analysis for the diagnostic efficacy of fibronectin in peritoneal fluid and plasma for endometriosis. The area under the curve (AUC) for fibronectin in peritoneal fluid was 0.932 (95% CI: 0.872–0.992, SD = 0.03) and 0.728 (95%CI: 0.607–0.848, SD = 0.06) for plasma. The estimated cut-off values for fibronectin as a biomarker of endometriosis equalled 13.15 [μg/L] for peritoneal fluid and 320.1 [mg/L] for plasma (Table 5). The results of combined efficacy of the plasma and peritoneal fibronectin are as follows: AUC = 0.993, 95%CI: 0.981–1.00, SD = 0.006, *p* < 0.001). Estimated sensitivity and specificity are presented in Table 6. The results of binary logistic regression to predicted probability for both parameters were SS model 9.29 and MS model 4.64, df = 2, *p* = 0.001.

## 3. Discussion

The main finding of this study is the observation that patients with endometriosis, compared to healthy controls, have elevated levels of fibronectin both in the peritoneal fluid and plasma. A strong statistical significance was observed for both specimens.

Lis-Kuberka et al. observed significantly higher concentration of fibronectin in the blood plasma (292.61 ± 96.17 mg/L vs. 226.55 ± 91.98 mg/L in controls) of women with endometriosis and the presence of fibronectin–fibrin complexes with a molecular mass of more than 1300 kDa, whereas there was a complete absence of these complexes in healthy women [39]. One possible explanation is the increased and chronic activation of coagulation mechanisms in patients with endometriosis. Another study that analysed eutopic endometrial tissue samples suggested the potential use of fibronectin as a clinical biomarker for detecting endometriosis [38]. We found only one study from 1988, performed on 22 women with endometriosis, which suggested a lower peritoneal fluid concentration of fibronectin [40]. Further studies suggest that fibronectin alone, as well as single nucleotide polymorphisms of fibronectin 1, may be involved in the pathogenesis of this disease [41]. Increased fibronectin gene expression indicated tissue injury in endometriosis compared to normal surrounding tissue [42]. Belard et al. suggested that fibronectin receptors could play a role in the persistence of endometriotic lesions despite menstruation in the corresponding eutopic endometrium [43].

We did not observe any significant differences in the concentration of collagen IV in peritoneal fluid or plasma. The hypothesised role of collagen IV in the pathogenesis of endometriosis is due to the interactions of T cells with ECM proteins, which lead to inappropriate proliferation and apoptosis of endometriotic implants [44]. Endometriosis has some similarities with the neoplastic process. It has been shown that a number of mutations in the endometrial epithelium can predispose cells to invasion and survival in the ectopic environment [45]. Lindgren et al. proposed type IV collagen as a promising biomarker for metastatic breast cancer. The authors of the study showed significantly increased protein levels in patients with metastases compared to those with primary tumours [(192 ng/mL (89.1–1395.9) vs. 73.6 ng/mL (44.6–187.7)]. Interestingly, the level of collagen IV was lower in patients with primary tumour, compared to the control group. These results suggest a role for collagen in the stage of cell invasion and not in the neoplastic transformation itself [46]. Another study examined the molecular mechanisms regulating the invasiveness of endometrial cells, where ECM proteins play a major role. In the present study, no differences were observed in the expression of collagen IV in the ectopic tissue or in the secretory endometrium [47].

Elevated concentrations of fibronectin do not seem to correlate with endometriosis severity. This is the first study to evaluate such a dependency. Among the samples from the study group, we did not observe any significant differences in fibronectin and collagen IV concentrations depending on the severity of endometriosis. A simple measurement of plasma fibronectin levels may be useful at each stage of endometriosis. It seems to be both a strength and weakness at the same time. Measurement of plasma fibronectin concentration could be equally useful for minimal and severe endometriosis; however, it does not differentiate the severity of the disease. Moreover, measurement of fibronectin in the peritoneal fluid may be useful for detecting endometriosis during laparoscopy. If validated, this might be helpful in patients with visually invisible lesions. Moreover, in multiple logistic regression, none of the assessed factors (patient’s age, phase of the menstrual cycle, or coexisting fertility impairments) affected the concentration of fibronectin in the peritoneal fluid or in the plasma. Therefore, we hypothesised that fibronectin is a reliable predictor of endometriosis because, apart from its high sensitivity and specificity, it is independent of individual variables.

The authors agree that there are several limitations to this study. First, the sample size of the study and the control group (49 patients with endometriosis and 35 controls) was defined by the availability of specimens obtained from the particular patient as well as funds obtained for further measurements. However, to the best of our knowledge, the present study had the largest sample size of all studies available in the literature that investigated the utility of fibronectin as a clinical biomarker of endometriosis. Future efforts related to this project will focus on extending the study population. We hope that our results will be published in the future. Other concerns arise from the nature of the cross-sectional study, the established criteria for patient enrolment, and the heterogeneity of the control group. Moreover, lead time bias could impact the results regarding the time of final diagnosis of endometriosis. This may result from the accidental detection of endometriosis during the infertility workup in previously asymptomatic patients.

The main strength of the study was the homogeneity of the study group, which was achieved due to rigorous exclusion criteria that allowed us to reduce the risk of bias. Moreover, we investigated potential biomarkers of endometriosis in both feasible plasma and peritoneal fluid collected during operative procedures.

Further studies with a larger sample size, evaluating the utility of plasma and peritoneal fibronectin levels alone or together with other biomarkers of endometriosis, should be carried out before they are introduced to common practice.

## 4. Materials and Methods

### 4.1. Study Populatoin

The research group was recruited from a cohort of a multicentre project throughout Poland (8 clinical centres, Department of Obstetrics and Gynaecology, Medical University of Warsaw; Angelius Provita Hospital in Katowice; Department of Gynaecology, Division of Infertility and Reproductive Endocrinology, Obstetrics and Gynaecological Oncology at Poznan University of Medical Sciences; Department of Obstetrics and Gynaecology, Central Clinical Hospital of the Ministry of Interior in Warsaw; Clinic of Obstetrics and Gynaecology, Provincial Combined Hospital in Kielce; Department of Surgical Gynaecology and Oncology, Medical University of Lodz; Department of Gynaecology and Obstetrics, Provincial Hospital in Przemysl; Department of Gynaecology, Gynaecology Oncology and Obstetrics, Institute of Medical Sciences, Medical College of Rzeszow University, grant no. 6/6/4/1/NPZ/2017/1210/1352). This cross-sectional study included women between 18 and 40 years of age who were qualified for planned laparoscopic surgeries due to one or more non-malignant conditions: infertility, chronic pelvic pain syndrome, ovarian cysts, and suspicion of endometriosis. The exclusion criteria were neoplasms, uterine fibroids, uterine septum, hormone therapy within three months preceding laparoscopy, pelvic inflammatory disease, irregular menstruation, and polycystic ovary syndrome. The control group comprised patients who were not diagnosed with endometriosis during the laparoscopic workup. The inclusion and exclusion criteria were the same in both groups. The sample size of the study and control groups was limited by the availability of information for individual statistical models, the number of all specimens taken from a particular patient, and the funds obtained for further measurements.

After providing written informed consent, all the patients underwent laparoscopic surgery. All surgeries were performed during the follicular phase of the menstrual cycle. Based on the findings of endometrial lesions assessed by the WERF EPHect Minimal Surgery Form (EPHect MSF) and confirmed by histopathology, the women were divided into study (endometriosis) and control groups. The severity of endometriosis was classified according to the recommendations of the American Society for Reproductive Medicine (rASRM) [48].

Personal data and baseline characteristics of individuals were obtained using a detailed questionnaire. A large part of the survey consisted of questions about pain, broken down by severity depending on the menstrual cycle, age, or sexual intercourse.

Prior to the surgery, blood samples were collected and stored in ethylenediaminetetraacetic acid (EDTA) 10 mL tubes (Sarstedt) in order to check plasma concentrations of the investigated biomarkers. Peritoneal fluid was aspirated at the beginning of the laparoscopy via a Veress needle under direct visual inspection to avoid contamination with blood. Each time, the procedure was performed in accordance with the Endometriosis Phenome and Biobanking Harmonisation Project standardisation. Material collection did not have any impact on the medical management of the patients and was performed in compliance with the Declaration of Helsinki. The aspirated peritoneal fluid was centrifuged at 1000 rpm for 10 min at 4 °C. The supernatant was transferred to a fresh 10 mL tube (Sarstedt). The same types of tubes were used for blood and peritoneal fluid collection in all centres included in the study. The time lapse between sample collection (both peritoneal fluid and plasma) and processing was less than 45 min. All centres centrifuged blood samples at 2500× *g* for 10 min at 4 °C. Subsequently, all specimen samples were stored at −80 °C until further measurements.

### 4.2. Reagents

Fibronectin from human plasma (lyophilised powder) as a standard, anti-fibronectin antibody produced in rabbit, collagen type IV, purified monoclonal mouse anti-human collagen type IV (Tebu-bio, Le Perray-en-Yvelines, France), bovine serum albumin (BSA), cysteamine hydrochloride, N-ethyl-N’-(3-dimethylaminopropyl) carbodiimide (EDC) (Sigma-Aldrich, Steinheim, Germany), N-hydroxysuccinimide (NHS), (Sigma-Aldrich, Munich, Germany), photopolimer ELPEMER SD 2054, and hydrophobic protective paint SD 2368UV SG-DG (PETERS, Kempen, Germany) were used, as well as absolute ethanol, acetic acid, hydrochloric acid, sodium hydroxide, sodium chloride, sodium carbonate, sodium acetate, (POCh, Gliwice, Poland). HBS-ES buffer pH = 7.4 (0.01 M HEPES, 0.15 M sodium chloride, 0.005% Tween 20, 3 mM EDTA), Phosphate Buffered Saline (PBS) pH = 7.4, carbonate buffer pH = 8.5 (BIOMED, Lublin, Poland, http://www.biomed.lublin.pl) were used as received. Aqueous solutions were prepared using Milli-Q water (Simplicity^®^ Millipore).

### 4.3. Procedures

#### 4.3.1. Chip Preparation

Biosensors for fibronectin and collagen IV determination were prepared as previously described [49,50]. The glass chips were covered with gold (50 nm on a 1 nm thick chromium layer). Next, the gold surface was covered with photopolymer and hydrophobic paint. An array of 9 × 12 free-gold surfaces was fabricated. Nine different solutions were applied simultaneously to the chip. Twelve independent measurements were carried out for each solution, so twelve individual SPRi signals for each sample were obtained.

#### 4.3.2. Antibody Immobilization

The chip surface was rinsed with absolute ethanol and water, dried in a stream of argon, and immersed in 20 mM cysteamine ethanolic solution for a minimum of 18 h. The chips with the immobilised linkers were rinsed and dried as described above. To immobilise the antibody, 50 µL of antibody solution (4 µg mL^−1^ for fibronectin determination and 6 µg mL^−1^ for collagen IV, designated from the experiments for optimisation of antibody concentration) was mixed with 250 µL of 50 mM N-hydroxysuccinimide (NHS), 250 µL of 200 mM N-ethyl-N’-(3-dimethylaminopropyl) carbodiimide (EDC), and 100 µL of carbonate buffer (pH = 8.5) and placed on the amine-modified surface. The chip was prepared in this manner and incubated at 37 °C for 1h. Afterwards, the surface of the prepared biosensor was rinsed several times with water and HBS-ES buffer pH = 7.40 (0.01 M HEPES, 0.15 M sodium chloride, 0.005% Tween-20, 3 mM EDTA).

#### 4.3.3. SPRi Measurement

Quantification of fibronectin and collagen IV was carried out using biosensors coupled with the SPR imaging technique, which was used as a detection method in earlier measurements [51]. The biosensor preparation is described in the preceding paragraphs.

SPRi measurements were performed in two polarisations. The p polarisation was used to observe changes in the intensity of active sites of the biosensor after binding of successive layers of the biosensor, whereas the s polarisation was the source of interference from the optical system of the device, which should be appropriately taken into account and corrected for changes in the intensity of the beam of radiation-exciting plasmons depending on the position in which the arm, on which the laser diode is placed, was located. The analytical signals obtained were then read using ImageJ software (National Institutes of Health, NIH), which led to the concentration values (after taking dilutions into account) of the given protein in the studied sample.

All the measurements were performed under stationary conditions. The contrast values obtained for all pixels across a single sample spot were integrated. A background correction was applied, that is, some of the areas on the biosensor covered with PBS buffer were used as a control. Non-specific binding was monitored by measuring the SPRi signal in the chip area that did not contain the receptor (ligand). Non-specific binding was minimised by preparing samples in PBS buffer and by placing BSA in PBS buffer on the chip. The SPRi signal, which was proportional to the mass of entrapped fibronectin or collagen IV, was obtained as the difference between the signals before and after interaction with the analysed sample for each spot separately.

### 4.4. Statistical Analyses

Descriptive statistics are presented as frequencies and percentages. The main differences were assessed after dividing the study population into two subgroups depending on whether endometriosis was confirmed during surgery or not. Student’s *t*-test and Pearson’s chi-square test were used to compare these groups in relation to fibronectin and collagen IV plasma and peritoneal fluid concentrations. The Mann–Whitney U test was used to assess the differences between the subsequent stages of endometriosis. One-way ANOVA (Kruskal–Wallis or Welch’s test) was used to assess the dependency between fibronectin and collagen IV concentrations and the severity of endometriosis. When interpreting results presented in larger contingency tables, we analysed adjusted standardised residuals, where the absolute value >1.96 corresponds to a significance of *p* < 0.05. In the multivariate analysis, a multinomial logistic regression model was estimated, with no endometriotic lesions as the reference category. For logistic regression, incomplete records were removed, which limited the file to 59 samples (36 with endometriosis and 23 controls). OR with 95% CI and *p*-value for each predictor considered in the paper was calculated. In order to determine the best regression model, multiple regression using the LR (Backward/Likelihood ratio) method was used. ROC curve analyses were performed for the selected models. Statistical software Statistica, v. 12.6 (TIBCO Software Inc., Palo Alto, CA, USA) was used for analysis.

## 5. Conclusions

SPRi biosensors have been successfully used for the determination of plasma fibronectin and collagen type IV as label-free methods for quantitative analysis. Therefore, plasma fibronectin levels should be considered a non-invasive marker of endometriosis. It may be useful at any stage of endometriosis because elevated concentrations of fibronectin do not correlate with the severity of the underlying disease. Assessment of fibronectin levels in peritoneal fluid could enhance the appropriate detection of disease during laparoscopy, especially in patients without visible endometriotic lesions. At present, plasma and peritoneal fluid collagen IV measurements do not appear to play a significant role as non-invasive markers of endometriosis.

## Figures and Tables

**Figure 1 ijms-23-15669-f001:**
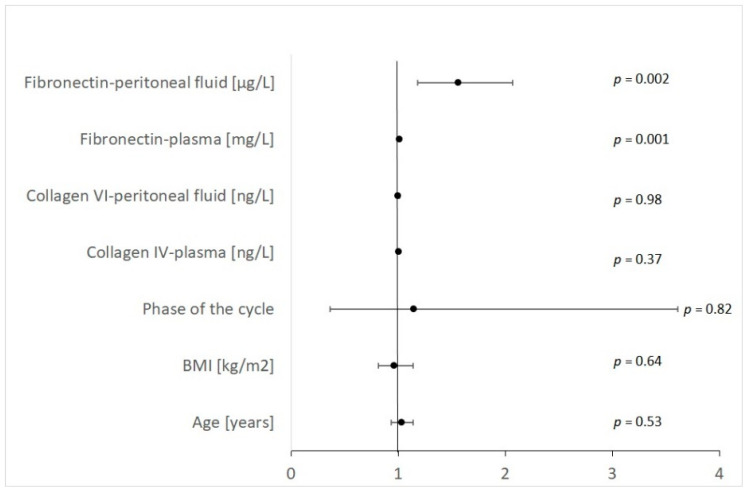
Logistic regression for endometriosis OR and 95% CI (n = 59).

**Figure 2 ijms-23-15669-f002:**
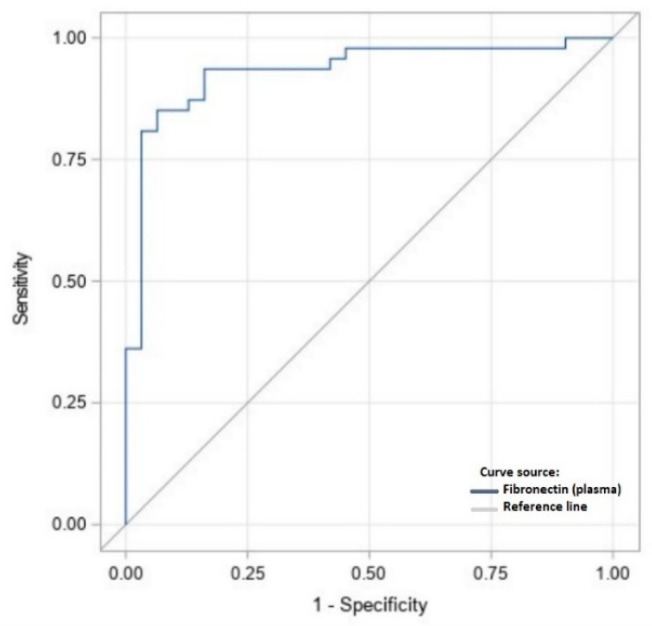
The ROC curve analysis for diagnostic efficacy of fibronectin in peritoneal fluid for endometriosis.

**Figure 3 ijms-23-15669-f003:**
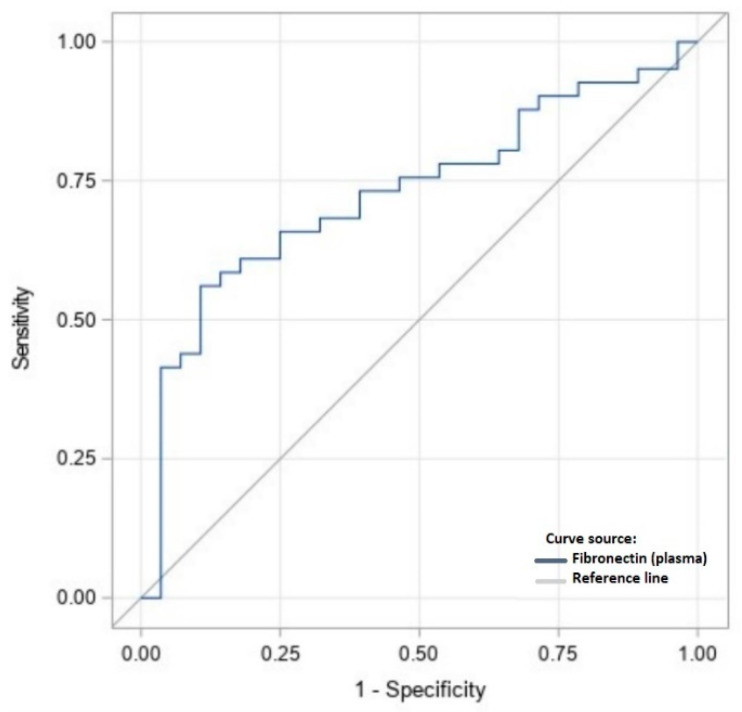
The ROC curve analysis for diagnostic efficacy of fibronectin in plasma for endometriosis.

**Figure 4 ijms-23-15669-f004:**
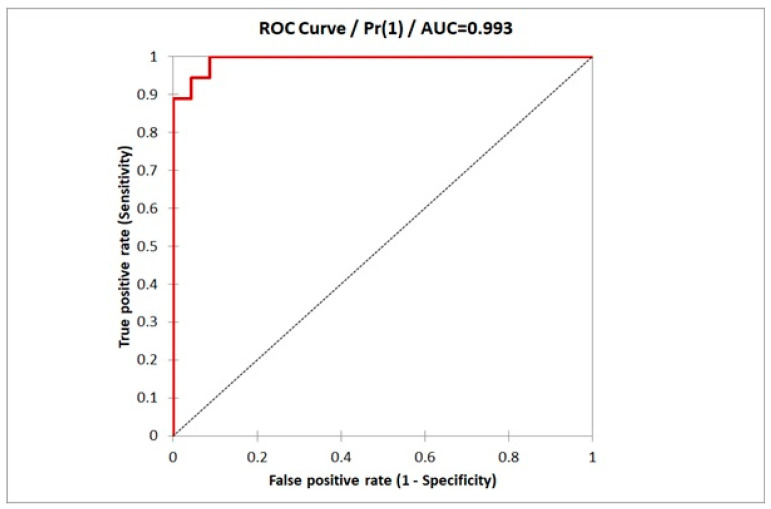
The ROC curve analysis for diagnostic efficacy of both fibronectin in plasma and peritoneal fluid for endometriosis.

**Table 1 ijms-23-15669-t001:** Baseline characteristics of the studied groups. The given values are the arithmetic mean. SD—standard deviation.

Variable	Study Group, n = 49 (SD)	Control Group, n = 35 (SD)	*p*-Value
Age [years] n = 74	32 (±4.8) n = 46	31.3 (±5.9) n = 28	0.55
BMI [kg/m^2^] n = 70	22.3 (±2.8) n = 45	22.4 (±3.7) n = 25	0.97
Any gestation in the past n = 72	0.19 (±0.40) n = 46	0.31 (±0.47) n = 26	0.29

**Table 2 ijms-23-15669-t002:** Plasma and peritoneal fluid concentrations of fibronectin and collagen IV. SD—standard deviation.

Biomarker	Study Group (SD)	Control Group (SD)	*p*-Value
Fibronectin plasma [mg/L] n = 69	329.3 (±98.5) n = 41	251.2 (±84.0) n = 28	0.001
Fibronectin peritoneal fluid [μg/L] n = 78	26.8 (±11.1) n = 47	7.0 (±5.9) n = 31	<0.001
Collagen IV plasma [ng/L] n = 69	559.1 (±88.6) n = 41	540.6 (±82.6) n = 28	0.385
Collagen IV peritoneal fluid [ng/L] n = 76	572.6 (±72.0) n = 45	583.6 (±50.9) n = 31	0.465

**Table 3 ijms-23-15669-t003:** Fibronectin and collagen IV concentrations depending on the stage of endometriosis. SD–standard deviation, one-way ANOVA (Kruskal–Wallis or Welch’s test).

Biomarker	I (SD)	II (SD)	III (SD)	IV (SD)	*p*-Value
Fibronectin plasma [mg/L] n = 41	318.0 (±102.6) n = 10	326.5 (±121.3) n = 7	341.1 (±76.4) n = 15	324.1 (±122.3) n = 9	0.029
Fibronectin peritoneal fluid [μg/L] n = 47	25.2 (±13.3) n = 13	27.4 (±11.5) n = 7	29.5 (±8.6) n = 18	22.9 (±12.5) n = 9	0.001
Collagen IV plasma [ng/L] n = 41	539.8 (±99.7) n = 10	568.2 (±81.9) n = 7	562.1 (±68.9) n = 15	568.6 (±118.6) n = 9	0.702
Collagen IV peritoneal fluid [ng/L] n = 45	595.4 (±58.6) n = 13	582.2 (±124.0) n = 7	575.3 (±49.0) n = 16	527.2 (±66.8) n = 9	0.109

**Table 4 ijms-23-15669-t004:** Logistic regression for endometriosis OR and 95% CI (n = 59).

Variable	OR	−95%CI	+95%CI	Wald Chi-Square	*p*
Age [years]	1.03	0.93	1.14	0.40	0.528
BMI [kg/m^2^]	0.96	0.81	1.14	0.22	0.638
Phase of the cycle	1.14	0.36	3.61	0.05	0.816
Collagen IV-plasma [ng/L]	1.00	1.00	1.01	0.82	0.366
Collagen VI-peritoneal fluid [ng/L]	1.00	0.99	1.01	0.00	0.983
Fibronectin—plasma [mg/L]	1.01	1.01	1.02	11.06	0.001
Fibronectin—peritoneal fluid [μg/L]	1.56	1.18	2.07	9.99	0.002

**Table 5 ijms-23-15669-t005:** Suggested cut-off values and estimated specificity and sensitivity of plasma and peritoneal fluid fibronectin.

Variable	Plasma [mg/L]	Peritoneal Fluid [μg/L]
Suggested cut-off value	320.1	13.15
Specificity	0.73	0.93
Sensitivity	0.61	0.87

**Table 6 ijms-23-15669-t006:** Estimated sensitivity and specificity for combined model of the plasma and peritoneal fibronectin.

Sensitivity	−95% CI	+95 CI	Specificity	−95% CI	+95 CI
1.000	0.882	1.000	0.913	0.718	0.986

## Data Availability

The data presented in this study are available upon request from the corresponding author. The data were not publicly available because of privacy concerns.

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
