# Peer review of "Plasma and Peritoneal Fluid Fibronectin and Collagen IV Levels as Potential Biomarkers of Endometriosis"

_ijms, 2022, doi:10.3390/ijms232415669_

Round 1

Reviewer 1 Report

In this study, the authors investigated fibronectin and type IV collagen in peritoneal fluid and plasma as potential biomarkers in the diagnosis of endometriosis. I have several concerns to be addressed as follows:

1-     Introduction:

-        The epidemiological data for endometriosis incidence should be supported by a recent reference. Thus, Reference 1 should be updated.

-        More literature review on the SPRI technique and its principles.

-        The hypothesis should be defined in relation to the controversy among previous studies on fibronectin and the limited data on type IV collagen.

2-     Materials and methods:

-        The criteria upon which endometriosis severity was determined should be supported by a reference.

-        Statistical analysis: Logistic regression was mentioned as being used. However, no data for these tests were presented in the results.

3-     Results:

-        The sample size is too small, which is further reduced upon subgrouping according to severity. Please add justification for the sample size used.

-        The control group should be defined for the included disorders.

-        In table 2, the measurement unit of fibronectin and type IV collagen should be defined. The results of fibronectin in peritoneal fluid appear to be non-normally distributed. Please check for normality.

-        Logistic regression analysis should be performed for fibronectin and type IV collagen in peritoneal fluid and plasma for the prediction of endometriosis.

-        The diagnostic efficacy of fibronectin in peritoneal fluid and plasma for endometriosis should be assessed using the Roc curve analysis with the combined efficacy of the plasma and peritoneal fibronectin too.

-        Correlation between plasma and peritoneal levels of fibronectin in the endometriosis group should be assessed.

4-     Discussion: There are several limitations that should be acknowledged, such as lead-time bias and heterogeneity of the control group.

5-     There are several structural and grammatical errors that need careful revision for the English language.

Author Response

Dear Reviewer,

We would like to thank you for taking the necessary time and effort to review the manuscript. We have carefully considered the comments. We sincerely appreciate all your valuable comments and suggestions, which helped us in improving the quality of the manuscript.

Below we enclose point-by point response:

1-     Introduction:

- The epidemiological data for endometriosis incidence should be supported by a recent reference. Thus, Reference 1 should be updated.

- It has been appropriately changed. We have updated these reference.

Smolarz, B.; Szyllo, K.; Romanowicz, H. Endometriosis: Epidemiology, Classification, Pathogenesis, Treatment and Genetics (Review of Literature). Int J Mol Sci 2021, 22, doi:10.3390/ijms221910554. (Line 51.*)

- More literature review on the SPRI technique and its principles.

- To highlight the innovative SPRI method, we've added additional information on how it works.

“It involves the application of the phenomenon of coupling light energy with an electromagnetic field on a metal surface plasmon resonance imaging of cells and surface-associated fibronectin. It is characterized by a very sensitive measurement of the refractive index, which makes it ideal for the quantification of molecules such as proteins (Surface Plasmon Resonance Imaging to Probe Dynamic Interactions Between Cells and Extracellular Matrix) [33]. An additional advantage of this innovative technique is that it does not require any markers [35]”.  (Lines 97-103, 106-107.)

- The hypothesis should be defined in relation to the controversy among previous studies on fibronectin and the limited data on type IV collagen.

- As suggested, we highlighted it in the lines 109-112 and supported it with appropriate references.

“To date, type IV collagen has not been tested for its usefulness in detecting endometriosis; however, few data indicate an increased adhesion of endometrial cells to this protein [37]. On the other hand, research on fibronectin has so far yielded inconclusive results, therefore further trials on larger groups of patients are necessary [38].”

2-     Materials and methods:

- The criteria upon which endometriosis severity was determined should be supported by a reference.

- As suggested, we added relevant reference.

“Lee, S.Y.; Koo, Y.J.; Lee, D.H. Classification of endometriosis. Yeungnam Univ J Med 2021, 38, 10-18, doi:10.12701/yujm.2020.00444.”  (Line 263.)

-  Statistical analysis: Logistic regression was mentioned as being used. However, no data for these tests were presented in the results.

- We have included the results of logistic regression into the main text.

“In multiple logistic regression for fibronectin and type IV collagen in peritoneal fluid and plasma for the prediction of endometriosis any of assessed factors (age, phase of the cycle or coexisting fertility impairments) achieved statistical significance (p < 0.05 for each variable). Any of previously mentioned factors did not correlate with the peritoneal or plasma levels of fibronectin and type IV collagen. We did not observe any significant correlation between the plasma fibronectin concentration and pelvic pain scale.” (Lines 140-145.)

3-     Results:

- The sample size is too small, which is further reduced upon subgrouping according to severity. Please add justification for the sample size used.

- The sample size of the study group was defined by the availability of all specimens obtained from the particular patient as well as funds obtained for the further measurements. As suggested, we added justification for the sample size to the main text.

“The sample size of the study group was defined by the availability of all specimens obtained from the particular patient as well as funds obtained for the further measurements”. (Lines 254-256.)

- The control group should be defined for the included disorders.

-  As suggested, we have defined control group for the included disorders.

“The exclusion criteria were: neoplasms, uterine fibroids, uterine septum and hormone therapy within three months preceding laparoscopy, pelvic inflammatory disease, irregular menstruations and polycystic ovary syndrome. The control group constituted the patients who were not diagnosed of endometriosis during laparoscopic workup. Inclusion and exclusion criteria were common for both groups.” (Lines 249-254.)

-  In table 2, the measurement unit of fibronectin and type IV collagen should be defined. The results of fibronectin in peritoneal fluid appear to be non-normally distributed. Please check for normality.

- As suggested, we have included the results of normality test. Table 2.

-  Logistic regression analysis should be performed for fibronectin and type IV collagen in peritoneal fluid and plasma for the prediction of endometriosis.

- We have included the results of logistic regression into the main text.

“In multiple logistic regression for fibronectin and type IV collagen in peritoneal fluid and plasma for the prediction of endometriosis any of assessed factors (age, phase of the cycle or coexisting fertility impairments) achieved statistical significance (p < 0.05 for each variable). Any of previously mentioned factors did not correlate with the peritoneal or plasma levels of fibronectin and type IV collagen. We did not observe any significant correlation between the plasma fibronectin concentration and pelvic pain scale.” Moreover in multiple logistic regression any of assessed factors (patient’s age, phase of the menstrual cycle or coexisting fertility impairments) did not impact on the concentration of fibronectin in peritoneal fluid as well as in plasma”.  (Lines 140-145, 204-207.)

- The diagnostic efficacy of fibronectin in peritoneal fluid and plasma for endometriosis should be assessed using the Roc curve analysis with the combined efficacy of the plasma and peritoneal fibronectin too.

- As suggested, we have included diagnostic efficacy of fibronectin in peritoneal fluid and plasma for endometriosis.

“Figure 1. and Figure 2. present respectively the Roc curve analysis for diagnostic efficacy of fibronectin in peritoneal fluid and plasma for endometriosis. Area Under ROC Curve for fibronectin in peritoneal fluid equaled 0.932 (95% CI 0.872 to 0.992, SD = 0.03) and 0.728 (95%CI 0.607 to 0.848, SD = 0.06) for plasma. The estimated cutoff values for fibronectin as a biomarker of endometriosis equaled 17.1 [μg/L] for peritoneal fluid and 338.1 [mg/L] for plasma”. (Lines 146-151, Figures 1. and 2.)

4- Discussion: There are several limitations that should be acknowledged, such as lead-time bias and heterogeneity of the control group.

-As suggested, we have added limitations section into the manuscript.

“The authors agree that there are several limitations of our study. First is the sample size of the study and the control group (49 patients with endometriosis and 35 controls) that was defined by the availability specimens obtained from the particular patient as well as funds obtained for the further measurements. However, to the best our knowledge presented paper enrolled the largest sample size of all studies available in the literature that investigated the utility of fibronectin as clinical biomarker of endometriosis. In the future efforts related to our project will focus on extending study population. We hope that we could publish these results in the future. Some other concerns arise from the nature of the cross-sectional study and the established criteria for patients enrolment and heterogeneity of the control group. Moreover the lead-time bias could impact on the results regarding the time of final diagnosis of endometriosis. It may result from the accidentally detected endometriosis during infertility workup in previously asymptomatic patients”. (Lines 210-222.)

5- There are several structural and grammatical errors that need careful revision for the English language.

We made every effort to correct the errors. We have carefully revised the manuscript for the English language.

* The location of the changes is given for read mode enabled with change tracking. If it is switched off the changes have a different numbering.

Author Response

Dear Reviewer,

We would like to thank you for taking the necessary time and effort to review the manuscript. We have carefully considered the comments. We sincerely appreciate all your valuable comments and suggestions, which helped us in improving the quality of the manuscript.

Below we enclose point-by point response:

Major points:

- The authors should present novel evidence. According to the manuscript this study showed the same results as previously showed reports, in which the fibronectin concentration in blood plasma with endometriosis is higher than without endometriosis, and peritoneal samples showed the same result. I think the authors could emphasis the novel SPRI method to assess the fibronectin concentration.

- So far, the number of studies investigating fibronectin as a potential diagnostic tool in endometriosis has been limited. To the best of our knowledge, we believe we have covered the latest scientific reports on this topic in the Discussion paragraph.

“Lis-Kuberka et al. observed significantly higher concentration of fibronectin in the blood plasma (292.61 ± 96.17 mg/L vs 226.55 ± 91.98 mg/L in controls) of women with endometriosis and the presence of FN-fibrin complexes with a molecular mass of more than 1300 kDa, whereas there was a complete absence of these complexes in healthy women [39]. The possible explanation is an increased and chronic activation of coagulation mechanisms in patients suffering from endometriosis. Another study that analyzed eutopic endometrial tissue samples suggests a potential use of fibronectin as a clinical biomarker to detect endometriosis [38]. We found only one study from 1988, performed on 22 women with endometriosis that suggested lower peritoneal fluid concentration of fibronectin [40]. Further studies suggested that fibronectin alone, as well as single nucleotide polymorphisms of fibronectin 1 may be involved in the pathogenesis of this disease [41]. Increased fibronectin gene expression points towards tissue injury in endometriosis, as compared to the normal surrounding tissue [42]. Belard et al. suggested that fibronectin receptors could play a role in the persistence of endometriotic lesions, despite menstruation in corresponding eutopic endometrium [43].” (Lines 163-177.*)

We agree that the innovation of the chosen assay method has not been fully highlighted, so we have added an additional excerpt.

“It involves the application of the phenomenon of coupling light energy with an electromagnetic field on a metal surface plasmon resonance imaging of cells and surface-associated fibronectin. It is characterized by a very sensitive measurement of the refractive index, which makes it ideal for the quantification of molecules such as proteins (Surface Plasmon Resonance Imaging to Probe Dynamic Interactions Between Cells and Extracellular Matrix) [33].” “An additional advantage of this innovative technique is that it does not require any markers [35].” (Lines 97-103, 106-107.)

- Is there any correlation among the plasma or peritoneal fibronectin concentration? In case the clinical use, this should be clarified. Moreover, correlation between the plasma fibronectin concentration and pelvic pain scale should be demonstrated, because the authors could not find any differentiation among rASRM categories, which should be the most important question in this study.

- The suggested above statistics have been introduced.

“In multiple logistic regression for fibronectin and type IV collagen in peritoneal fluid and plasma for the prediction of endometriosis any of assessed factors (age, phase of the cycle or coexisting fertility impairments) achieved statistical significance (p < 0.05 for each variable). Any of previously mentioned factors did not correlate with the peritoneal or plasma levels of fibronectin and type IV collagen. We did not observe any significant correlation between the plasma fibronectin concentration and pelvic pain scale”. (Lines 140-145. Table 3.)

- If fibronectin is proposed as a biomarker, I thought that cutoff values should be calculated and sensitivity and specificity should be presented.

- Sensitivity and specificity of fibronectin as a biomarker of endometrioses have been introduced.

“Figure 1. and Figure 2. present respectively the Roc curve analysis for diagnostic efficacy of fibronectin in peritoneal fluid and plasma for endometriosis. Area Under ROC Curve for fibronectin in peritoneal fluid equaled 0.932 (95% CI 0.872 to 0.992, SD = 0.03) and 0.728 (95%CI 0.607 to 0.848, SD = 0.06) for plasma. The estimated cutoff values for fibronectin as a biomarker of endometriosis equaled 17.1 [μg/L] for peritoneal fluid and 338.1 [mg/L] for plasma”. (Lines 146-151. Figures 1. and 2.)

Minor points:

- In line 166, the authors cannot state that the peritoneal fluid sampling could improve the sensitivity in searching for invisible endometriosis, because the authors have not yet clarified  the sensitivity or specificity of the method in this study.

- This issue has been appropriately changed.

“On the other hand, measurement of fibronectin in the peritoneal fluid could potentially be useful for detecting endometriosis during laparoscopy. If validated it might be helpful in patients with visually invisible lesions”. (Line 202-204.)

- For the table 1, the authors should clarify the values as mean or median.

- This issue has been appropriately changed.

“Table 1. Baseline characteristics of the studied groups. The given values are the arithmetic mean. SD – standard deviation.” (Line 119.)

- For the table 3, if examine the 4 groups, one way Anova test (Kruskal-Wallis or Welch) should be used.

- This issue has been appropriately changed.

“Among the samples from the study group, we did not observe any significant differences between fibronectin and collagen IV concentrations depending on the severity of endometriosis. These data are presented in Table 3”.

“Table 3. Fibronectin and collagen IV concentrations depending on the stage of endometriosis. SD – standard deviation, on way Anova test (Kruskal-Wallis or Welch)”. (Table 3. Lines 134-138.)

- Page4, line 136: typo

- This issue has been appropriately changed. (change “lover” to lower” Line 171.)

Additional comments:

I thought the results of this study could be used clinically is to detect endometriosis to make preoperative assumptions about the difficulty of surgery (i.e., surgical time, blood loss, transfusion rate, or other organ injuries). For this purpose, I think the authors could include other benign ovarian tumors or uterine tumors (e.g., dermoid tumor, serous cyst, myoma, or adenomyosis) which were excluded this study, and I would like you to consider this in the future.

- In the future, after obtaining further fundings, efforts related to our project will focus on extending study population including patients with benign ovarian tumors or uterine tumors. We hope that we could publish these results in the future.

* The location of the changes is given for read mode enabled with change tracking. If it is switched off the changes have a different numbering.

Round 2

Reviewer 1 Report

The authors have adequately addressed some of my concerns, while others still to be addressed as follows:

1-     Results:

-        The results of logistic regression analysis should be tabulated.

-        The combined efficacy of the plasma and peritoneal fibronectin should be defined. In addition, the specificity and sensitivity of plasma and peritoneal fibronectin should be determined.

-        Correlation between plasma and peritoneal levels of fibronectin in the endometriosis group should be assessed.

2-     There are still several structural, grammatical, and typographical errors that need careful revision for the English language.

Round 3

Reviewer 1 Report

Dear authors,

I greatly appreciate your efforts in addressing my queries and concerns; however, the two points related to statistical analysis were inadequately addressed. The authors must seek the advice of a more professional statistician.

The logistic regression analysis should be conducted appropriately with endometriosis vs. control groups as dependent factors, with age, BMI, phase of the cycle, collagen IV-plasma, collagen VI-peritoneal fluid, fibronectin-plasma, and fibronectin-peritoneal fluid as covariates. This should be conducted as a univariate analysis with significant covariates entering the multivariate analysis and the results presented as an OR and 95% CI with p-values.

The combined efficacy of the plasma and peritoneal fibronectin was not appropriately conducted. They can use binary logistic regression analysis to estimate the predicted value (probabilities) and then ROC curve analysis to determine the diagnostic efficacy of both parameters in combination.

The correlation between plasma and peritoneal levels of fibronectin in the endometriosis group needs the correlation coefficient (r) to be included.

Round 4

Reviewer 1 Report

The authors have adequately addressed all my concerns and queries.

Author Response

We are glad that we could answer any doubts and questions.